# Ab-initio dynamic study of mechanisms for dust-mediated molecular hydrogen formation in space
Yuzhen Guo ⦿ ✉ & David R. McKenzie ⦿ ✉

The reason for the abundance of molecular hydrogen ($H_2$) in space remains unresolved. Here we study collision dynamics under spacelike conditions to test $H_2$ formation mechanisms where carbonaceous dust grains may have a catalytic role. Density functional theory molecular dynamics simulates atomic hydrogen capture and $H_2$ formation on the surface of buckminsterfullerene as a carbonaceous cosmic dust model. Maximally localized Wannier functions are applied to examine the electronic bonding during transition states. The fullerene surface is shown to be effective at warm (50K) and low (10K) temperatures in achieving atomic H chemisorption, potentially explaining the observed broad temperature range for efficient $H_2$ formation. We revise the Eley-Rideal mechanism and propose that both it and the Langmuir-Hinshelwood mechanism, induced by thermal hopping, contribute to bursts of $H_2$ formation during energetic events. Additionally, we show how fullerene maintains the abundance of $H_2$ in space by selectively preventing $H_2$ molecules from capture.

Molecular hydrogen ($H_2$) is the most abundant molecule in the interstellar medium[1] and dominates the mass budget of gas in regions of star formation. The presence of hydrogen in molecular form is vital in astrophysics because it not only contributes to the initial collapse of molecular clouds that leads to star formation[2], but also because it contributes to the synthesis of complex molecules in the interstellar medium[2]. However, the detailed formation mechanism for molecular hydrogen in interstellar clouds remains an open question given the probability for forming molecular hydrogen by two body collisions of atomic hydrogen in space is negligible. The low temperatures and low density that generally apply in interstellar space make two body atomic hydrogen collisions improbable. Besides that, the dissipation of the high recombination energy of the exothermic reaction $H + H \rightarrow H_2$ combined with the initial kinetic energy is not possible, leading to immediate re-dissociation[3]. It is now widely believed that catalytic reactions on the surfaces of dust grains in the interstellar medium are responsible for the formation of $H_2$ and other more complex molecules[4]. The correlation between molecular-to-atomic ratio of hydrogen and metallicity adds support to this idea[5].

Dust grains could act as catalysts by offering a surface where absorbed hydrogen atoms can associate and react to form molecules, with the grains absorbing the energy released, preventing immediate dissociation of the newly released $H_2$. Two main mechanisms have been proposed: the Langmuir-Hinshelwood (LH) mechanism and the Eley-Rideal (ER) mechanism. In the LH mechanism, it was proposed that H atoms physically adsorbed on the surface of the grain can migrate from one adsorption site to the other, and form $H_2$ when they encounter another H atom adsorbed on the surface; while in the ER mechanism, $H_2$ is formed by direct recombination of one incoming H atom with another H atom already trapped on the surface. Experimental investigation demonstrates the catalytic role of carbon, silicate, and water ice at temperatures less than 20K[6–8]. In addition, observational evidence indicates that $H_2$ can be efficiently formed over a wide range of temperatures[9,10]. The LH mechanism operating by physisorption is believed to dominate $H_2$ formation at low temperatures around 10K[11]. How molecular hydrogen forms on dust grains at higher temperature (20K < T < 100K) has been identified as an important unanswered question[12]. The formation of $H_2$ via energetic events, such as UV irradiation of hydrogenated carbons, astrophysical shocks and the interaction of high temperature gas with cold surfaces, is also a route that has been recommended for detailed investigation[11].

The composition of interstellar dust is inferred from the observed abundances in the gas phase, with elements that are significantly depleted considered to be present in condensed matter[12]. Silicate and carbon are the primary composition of dust grains[13]. While silicate grains are generally assumed to be glassy or amorphous, carbonaceous dust can exist in various forms, such as amorphous, graphitic and fullerene structures[12]. A previous study on carbon dust suggests that the formation of molecular hydrogen is likely to take place on surface sites with defects that lead to fullerene-like

Materials Physics Laboratory, School of Physics, The University of Sydney, Physics Rd, Camperdown, Sydney, NSW, 2006, Australia.
✉e-mail: yguo8229@uni.sydney.edu.au; david.mckenzie@sydney.edu.au

curvature[3]. Even in graphene or amorphous carbon structures, curved sheets exhibit a strong local resemblance to fullerene[14]. In our study, we model the surface of fullerene $C_{60}$ as a representative of larger carbonaceous dust grains, although the presence of free $C_{60}$ is also plausible.

The detection of fullerene in unidentified infrared emission (UIE) bands and diffuse interstellar bands (DIBs) suggests that fullerenes are abundant in the interstellar medium. UIE bands are discrete infrared emissions originating from circumstellar envelopes and the interstellar medium, for which the identity of the emitting species is unknown. Most of the UIE bands have been associated with vibrational modes of $sp^2$ and $sp^3$ C-H and C-C bonds[15–17]. While the exact origin of such bonds remains unknown, it has been speculated that fullerene and fullerene-related species are abundant in the interstellar medium and contribute to part of the UIE bands. The UIE bands at 7.0, 8.5, 17.4 and 18.9 μm, match the four vibrational transitions of $C_{60}$, providing evidence for the existence of $C_{60}$ in interstellar dust[18,19]. The study of diffuse interstellar bands (DIBs) further supports the existence of fullerene in space. Up to the present time, many DIBs have been detected at ultraviolet, visible, and infrared wavelengths. The stable structure and abundance of fullerene molecules make them ideal candidates as DIB carriers. While several claims have been made over many years that specific DIBs could be assigned to specific species, the Buckminsterfullerene cation ($C_{60}^+$), was the first, and is so far the only, molecule with a spectrum remarkably consistent with several DIBs[20–22]. Omont[22] investigated the basic interstellar properties and the likely distribution of different fullerene molecules, and concluded that the general landscape of interstellar fullerene compounds is probably much richer than previously realized.

Some researchers also suggest that the existence of astronomical hydrogenated fullerenes (fulleranes) in space is likely. The detection of fullerenes in a series of H-containing planetary nebulae[23] shows that fullerenes exist (and may be formed) in H-containing environments. Experimental studies of mixing hydrogen atoms and $C_{60}$ in solvents demonstrated that $C_{60}$ can be easily hydrogenated into $C_{60}H_{36}$ by atomic hydrogen[24,25]. On the other hand, the heating of $C_{60}H_{36}$ in laboratory conditions shows an efficient dehydrogenation with release of $H_2$ molecules[24,25]. Stodt et al.[26] reported the production of mid-infrared features that match the UIE bands when they mix atomic hydrogen with $C_{60}$. The electronic absorption spectrum of $C_{60}H_{36}$ in the vicinity of 217 nm matches the observed spectrum of the interstellar extinction of light at 217.5 nm, with an absorption spectrum that also matches several UIE bands detected in protoplanetary nebulae[27]. Zhang et al.[28] presented a thorough discussion of the possible roles of fulleranes in DIBs, UIE, extended red emission, and anomalous microwave emission. These observations support the idea that fullerenes may themselves play a key role in the catalytic formation of interstellar molecular hydrogen as well as being representative of the other types of carbonaceous dust grain that might participate in $H_2$ formation. Currently, most large-scale simulations of interstellar molecular hydrogen formation rely on rate or master equations[11]. However, understanding the detailed mechanisms involved is crucial for enhancing the accuracy of these approaches for future simulations that incorporate fullerenes.

In this paper, we use an ab-initio molecular dynamic scheme based on the quantum mechanical description of atom-atom interactions with the Kohn-Sham density functional method[29]. The density functional method is renowned for its precision in describing electron-mediated interactions without requiring prior assumptions of the hybridization state of orbitals in the participating atoms. It also excels in accurately describing systems containing carbon where mixed hybridization states co-exist[30]. The probability of collisional capture and the mechanism by which excess kinetic energy is dissipated after capture are many-body problems where the outcome is unknown without simulating the time evolution of the entire complex. Therefore, ab-initio molecular dynamics is a valuable tool. Questions that remain are: What is the energy dependence of the H-capture probability? Is it possible for the energy of an incoming H atom to be dissipated without disrupting the $C_{60}$ structure and is molecular hydrogen released without dissociation by subsequent collisional or thermal

disruption of the H-$C_{60}$ complex? To answer these questions, we simulate the bombardment by both $H_I$ and $H_2$ onto the surface of neutral $C_{60}$. The formation process of $H_2$, either by a subsequent particle collision or by absorption of energy from photons, is explored using fullerane $C_{60}H_{36}$ as a platform. Wannier function analysis is applied to unambiguously interpret the bonding in both the capture of $H_I$ and the formation of $H_2$. We study the hydrogenation of $C_{60}$ by adjusting the velocity and angle of incidence of the hydrogen atom, as well as changing the impact site on the molecule to estimate the probability of effective hydrogenation.

## Results and discussion

To begin, we stabilized the structure of $C_{60}$ and $C_{60}H_{36}$ at 10K (Figs. S1 and S2), a commonly reported temperature of cold molecular clouds. Car-Parrinello molecular dynamics (CPMD)[31] simulation was performed by means of the CPMD code[32] to efficiently stabilize the structure. After that, an extra hydrogen atom or molecule was introduced into the system. Born-Oppenheimer molecular dynamics (BOMD) was applied during the collision. The velocity of the additional hydrogen atom/molecule was adjusted in simulated collisions with the fullerene/fullerane. The temperature ranges for neutral interstellar medium[33] is 10–20K, 50–100K, 6000–10,000K for molecular clouds, cold neutral medium, and warm neutral medium respectively. We calculated the root mean squared velocity of hydrogen atom/molecule in the temperature range of 10K–6000K given by:

$$v_{rms} = \sqrt{\frac{3RT}{M}}, \quad (1)$$

where $R$ is the universal gas constant, $T$ is the temperature, and $M$ is the molecular mass. We take $M = 1$ g/mol for hydrogen atom and $M = 2$ g/mol for hydrogen molecule during the simulation.

We next explored the ability of neutral fullerene to capture $H_I$ in space. A microcanonical ensemble (NVE) was applied to make the simulation close to the almost adiabatic situation in space. We initialize the simulation with the $H_I$ moving towards a carbon atom of the $C_{60}$, or the ring area of the surface, under different initialized velocity calculated by Eq. (1) to explore the ability for $C_{60}$ to capture $H_I$ in space. The probability of fullerene to capture $H_I$ is written by:

$$P = f_C p_C + f_R p_R, \quad (2)$$

where $f_C$ is the probability of a hydrogen atom having a temperature in the range that results in capture when it is incident on the carbon atom, while $f_R$ is the probability of a hydrogen atom having a temperature in the range that results in capture when it is incident on the ring area. The probability that $H_I$ collides with a carbon atom is approximated by $p_C = \frac{A_C}{A_{C60}}$, where $A_C$ is the sum of cross sections of carbon atoms with atomic radius equals 0.914 Å, and $A_{C_{60}}$ is the surface area of $C_{60}$, while $p_R = 1 - p_C$ is the probability that the collisions happens at the ring area. Figure S3 illustrates the capture of a hydrogen atom as a function of temperature. This can be applied to the calculation of $f_C$ and $f_R$ when the incident velocity of hydrogen atom is normal to the impact site. As the incident angle increases, the temperature range over which hydrogen atom can be captured also expands. This occurs because an increase in the incident angle decreases the velocity component normal to the impact site at a given temperature. The Supplementary Movies 6 and 7 demonstrate the capture of atomic hydrogen as the incident angle increases.

For temperatures of the incident $H_I$ within 10K–100K, which corresponds to the temperature of molecular clouds (10–20K) and cold neutral medium (50–100K), all $H_I$ are captured by the fullerene. It was previously believed that capture of a cold hydrogen atom at 10K on the dust grain is determined by physisorption because cold atoms do not have enough energy to overcome the potential barrier between physisorption and chemisorption. A study on hydrogen adsorption on graphene[34] suggests that the primary factor contributing to the chemisorption barrier is the

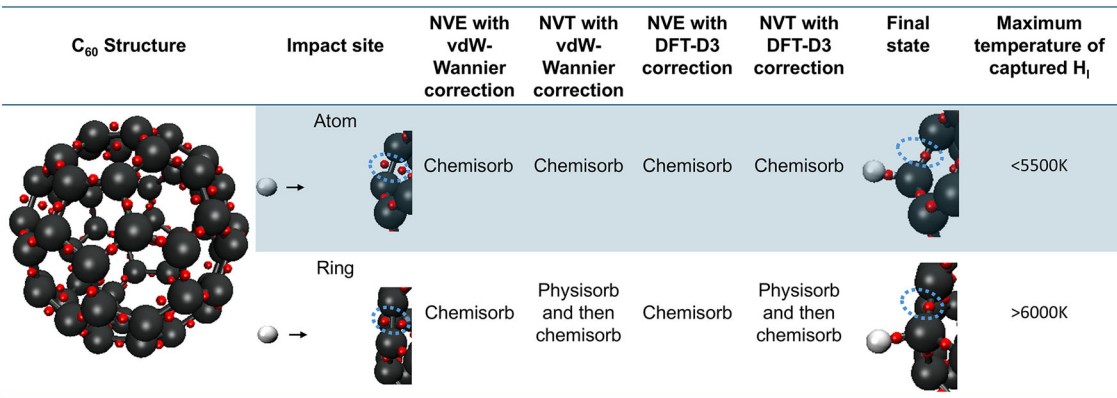

| $C_{60}$ Structure | Impact site | | NVE with vdW-Wannier correction | NVT with vdW-Wannier correction | NVE with DFT-D3 correction | NVT with DFT-D3 correction | Final state | Maximum temperature of captured $H_I$ |
|---|---|---|---|---|---|---|---|---|
| | Atom | | Chemisorb | Chemisorb | Chemisorb | Chemisorb | | <5500K |
| | Ring | | Chemisorb | Physisorb and then chemisorb | Chemisorb | Physisorb and then chemisorb | | >6000K |

**Fig. 1 | Capture of $H_I$ by $C_{60}$ using different levels of theory and various impact sites.** Chemisorption of $H_I$ is obtained at 10K for all cases. The black atoms indicate carbon atoms, the white atom indicates hydrogen, and the red balls indicate maximally localized Wannier centers. NVE denotes the microcanonical ensemble, and NVT denotes the canonical ensemble. The vdW-Wannier correction denotes the van der Waals correction proposed by Ambrosetti and Silvestrelli[41], and the DFT-D3 correction denotes van der Waals correction proposed by Grimme et al.[42]. The top row shows the capture of a hydrogen atom when the impact site is a $C_{60}$ atom. The bottom row shows the capture of hydrogen atom when the impact site is a ring center. The blue ring area shows the switch from a C-C double bond to a C-C single bond. The black arrow indicates the direction of the incident $H_I$. Heat is released during this process to assist the formation of C-H bond. The $H_I$ atom will directly pass through the physisorption site and enter the chemisorption site when the impact site is an atom of $C_{60}$. When the impact site is a ring area, together with a thermostat to maintain the temperature of the system at 10K, the $H_I$ atom will stay at the physisorption site for a short while before migrating to the chemisorption site. The final column shows the maximum temperature of $H_I$ being captured. $C_{60}$ fails to capture incoming $H_I$ with temperature of 5500K when the impact site is a carbon atom, while it can still capture $H_I$ with 6000K temperature when the impact site is a ring.

relaxation and rehybridisation of the adsorbent carbon atom directly beneath the H atom. Therefore, for a fullerene-like curved structure, it is reasonable to expect that chemisorption may still occur even at low temperatures, as the chemisorption barrier is lowered. Our simulation demonstrates the detailed dynamics of hydrogen atom chemisorption on the surface of fullerene at low temperatures (10K). When $H_I$ approaches a carbon atom, a C-C double bond changes to a C-C single bond, with energy released to help the hydrogen atom overcome the potential barrier between physisorption and chemisorption. A C-H bond is formed and then experiences a damped oscillation with the release of extra kinetic energy. When $H_I$ approaches the ring area of the fullerene, the same phenomenon is observed. The hydrogen atom is attracted to the carbon atom when their electronic clouds overlap, resulting in the transformation of a C-C double bond to a C-C single bond and the formation of C-H bond. It was confirmed that chemisorption dictates the outcome of the simulations at 10K and not physisorption by conducting additional simulations with van der Waals corrections included.

Overall, the simulations revealed that fullerene can effectively capture hydrogen atoms in both molecular clouds (Supplementary Movie 2) and the cold neutral medium (Supplementary Movie 8). Figure 1 summarizes the results. $H_I$ is observed to stay in the physisorption site for a while when the impact site is the ring area under the canonical ensemble (NVT), followed by migration to the chemisorption site to form a C–H bond (Supplementary Movie 1). The C–H bond is measured to be 1.11Å after stabilization, consistent with the experimental measurement of 1.09Å. This series of simulations demonstrates that hydrogen atoms can undergo chemisorption on the surface of $C_{60}$ even at 10K after initially being physisorbed. The fullerene fails to capture the hydrogen atoms that approach the carbon atom at a temperature of 5500K (Fig. S3 and Supplementary Movie 2). However, the initial kinetic energy of the atomic hydrogen is greatly reduced during the interaction, which makes it easily captured by any subsequent interaction with a fullerene. The fullerene can still capture incident $H_I$ at 6000K when it moves toward the ring area (Fig. S3 and Supplementary Movie 2). The probability of fullerene capturing $H_I$ is then approximated as unity in the molecular cloud and cold neutral medium under a relatively uniform and stable environment. The detailed dynamics as a function of temperature together with Eq. (2) make it possible to calculate more accurate hydrogen aggregation rate for large-scale simulations that depend on rate equations or master equations, enabling more reliable predictions.

The mechanism for the process to form molecular hydrogen was studied by the collision of the neutral hydrogen atom with a $C_{60}H_{36}$. The high level of hydrogenation in $C_{60}H_{36}$ was used to maximize the probability of a H-C-H collision. Temperatures of incoming hydrogen atoms were adjusted in the range from cold atoms (10K) to hot (6000K). The formation of hydrogen deuteride (HD) is also investigated by replacing the incident hydrogen atom by a deuterium. Figure 2 shows the separation of a 10K incoming $H_I$/D and an adsorbed hydrogen atom on $C_{60}H_{36}$ as a function of time. The increment of oscillation amplitude of the fullerane together with the transformation of $sp^3$ to $sp^2$ bond as indicated by the Wannier function centers, indicates the absorption of heat during the exothermic reaction $H + H \rightarrow H_2$ is accommodated by thermal vibration of the fullerene. Rapid oscillations of the H-H/H-D bond indicate a highly excited vibrational state of the nascent $H_2$/HD. Both $H_2$ and HD can be efficiently formed on the surface of carbonaceous grains for all initial temperatures of incident $H_I$ and D, provided that the impact site is not surrounded by three C-H bonds (Supplementary Movie 3). Otherwise, the hydrogen atom will remain near the surface, searching for sites where bond transformation can occur, or it may become physisorbed on the surface (Supplementary Movie 10). This suggests that, in addition to temperature, the local structure of the surface influences the formation efficiency of molecular hydrogen on the grain surface.

We have demonstrated the effectiveness for the formation of $H_2$ and HD on the surface of $C_{60}$ by the ER mechanism. As the temperature rises, the cage structure of fullerene allows it to maintain a stable structure, while the formation of a C-H bond ensures atomic hydrogen is adsorbed on the surface. Similar simulations (Supplementary Movie 9) conducted at a surface temperature of 50K have demonstrated that same mechanisms on the surface of fullerene also explain the formation of molecular hydrogen on a warm dust surface (20–100K). Both the capture of $H_I$ and the formation of $H_2$ on the surface release energy to the fullerene/fullerane. The radiative cooling rate of $C_{60}$ has been demonstrated to be two orders of magnitude higher than expected from infrared active vibrations[35]. Such efficient cooling would help to maintain the stable structure of fullerene during these processes. This heat sink function is likely to remain effective even if the fullerene-like structure forms part of a larger grain.

In astronomical environments, small grains can experience a sudden increase in temperature due to mechanisms such as shock waves and UV absorption. In this study, we explore the formation mechanisms of

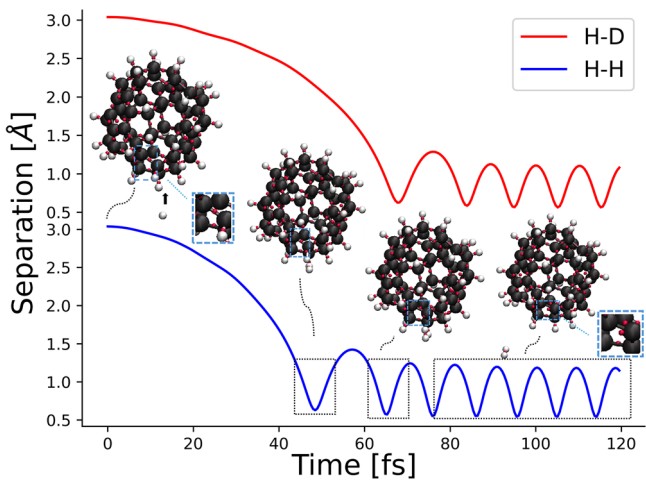

**Fig. 2 | The sequence of events showing the ER mechanism for HD and $H_2$ formation.** The red/blue line indicates the distance between the incident deuterium/hydrogen atom and the adsorbed hydrogen atom with which it collides as a function of time. The blue square region shows the switch from $sp^3$ to $sp^2$ bonding. The black arrow indicates the direction of movement of the incident $H_I$. The first minimum occurs during the collision of the incident $H_I$ with the adsorbed hydrogen atom on the fullerane. The second minimum shows the formation of hydrogen molecule by the ER mechanism, followed by the breakage of a C-H bond and the transformation of $sp^3$ bonding to $sp^2$ bonding. The rapid oscillation after formation indicates the highly excited vibrational state of the nascent molecular hydrogen. The amplitude of new formed $H_2$ is higher than that of HD. After 1 ps, the bond length of the $H_2$ is $0.90 \pm 0.22$ Å, while the bond length of HD is $0.87 \pm 0.19$ Å.

molecular hydrogen on the surface of dust during energetic events, using fullerane as a platform. The decomposition of fullerane ($C_{60}H_{36}$) was observed as a result of a sudden increase in temperature from 10K to 3000–4000K (Supplementary Movie 4). Unlike the "normal" ER mechanism described in Fig. 2, where molecular formation occurs through the collision of an incoming atomic hydrogen with an adsorbed atomic hydrogen, simulations reveal a revised ER mechanism takes place at high temperatures: the formation of molecular hydrogen by the recombination of a newly desorbed atomic hydrogen with an adsorbed hydrogen atom. Thermal hopping of atomic hydrogen on the surface with the formation of molecular hydrogen by the LH mechanism is now also observed. Figure 3 shows the formation of molecular hydrogen at 3000K by both the revised ER mechanism and the LH mechanism. When the temperature of the system rises to 4000K, a rapid decomposition of $C_{60}H_{36}$ is observed. A local hydrogen-dense area is formed during the decomposition, resulting in frequent collisions between hydrogen atoms and a more efficient formation of $H_2$ throughout the process. Similar simulations were carried out by replacing hydrogen atom with a heavier atom such as deuterium. The formation of molecular deuterium is observed under the same mechanism. This suggests catalytic formation on the surface of fullerene can be applied to explain the formation of both $H_2$ and $D_2$ in the interstellar medium. The mechanism under energetic events is distinct from both the "normal" ER and LH mechanisms, but would clearly dominate under the right conditions where a thermal shock event takes place.

Simulation of the collision with $H_2$ reveals another role of fullerene: the selective absorption of monoatomic hydrogen, which prevents molecular hydrogen from capture (Supplementary Movie 5). Hydrogen molecules are initialized at temperature 10K–6000K. In a similar way to the study of the impact of $H_I$ onto the surface of $C_{60}$, we investigated the collision of $H_2$ onto the carbon atom of the surface, or the ring area of the surface of $C_{60}$ using

**Fig. 3 | Thermal shock formation of a burst of molecular hydrogen using $C_{60}H_{36}$ as a model of a hydrogen covered dust grain.** A sudden increase of temperature from 10K to 3000K mimics an interaction with a UV photon or shock wave that delivers a temperature spike. The black atoms indicate carbon atoms, the white atoms indicate hydrogen atoms, and the red circles indicate hydrogen molecules. The transparent chain shows the trajectory during the formation of molecular hydrogen. The first hydrogen molecule is formed after 0.73 ps. The second and the third hydrogen molecules are formed after 2.39 ps and 4.81 ps, respectively. Both the ER mechanism and the LH mechanism induced by thermal hopping are observed during the process.

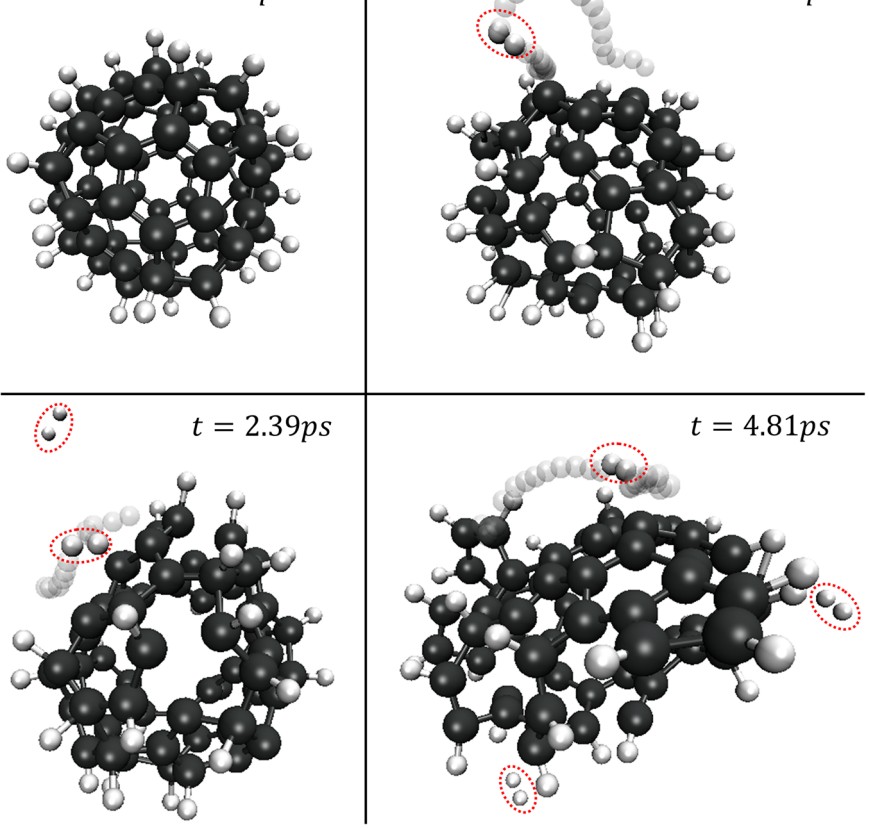

BOMD. Molecular hydrogen is prevented from reacting with $C_{60}$ under all temperatures and all incident directions, which ensures that molecular hydrogen does not decompose into atomic hydrogen on the surface of fullerene. This is expected as the dissociation energy of $H_2$ is extremely high[36]. Fullerene can therefore effectively convert atomic hydrogen into molecular hydrogen, while the reverse reaction is suppressed. This effect is also evident in the decomposition of fullerane, where atomic hydrogen that leaves the fullerane can easily re-enter the surface of the dust and form molecular hydrogen via the ER mechanism. The molecular hydrogens are then repelled after their formation.

In summary, this work demonstrates how dynamic phenomena occurring on the surface of carbonaceous dust can account for the efficient formation of molecular hydrogen across a broad temperature range. Our analysis, employing Wannier functions and incorporating van der Waals interactions, provides a high level of detail sufficient to discriminate between types of bond. We find that chemisorption of atomic hydrogen occurs on both warm (50K) and cold (10K) fullerene surface, while it was previously believed that physisorption dominates at 10K on dust surfaces. This mechanism suggests chemisorption can take place on a bare carbonaceous dust surface even at 10K. Although dust grains at low temperature are likely to be partially covered by an ice mantle, Potapov et al.[37] suggest that ice densities vary dramatically and the ice coverage of dust grains in cold astrophysical environments is less than has typically been assumed. This finding implies that chemisorption of atomic hydrogen on the surface of carbonaceous dust may also be a significant process at low temperatures, particularly during the phase when an ice mantle has not completely formed. Efficient capture of $H_I$ is observed over a wide range of $H_I$ temperatures, with fullerenes capable of capturing cold hydrogen atoms or cooling them from higher temperature to facilitate their subsequent capture. Our simulations illustrate that both the formation of $H_2$ and HD is efficient on the dust surface by the ER mechanism. Additionally, we demonstrate that not only temperature, but also the local structure of the grain surface, significantly impact the recombination efficiency of molecular hydrogen. The revised ER mechanism, together with the LH mechanism, contributes to $H_2/D_2$ formation during energetic events. Furthermore, we propose that fullerenes contribute to the abundance of $H_2$ in space through a preferential adsorption of atomic hydrogen relative to molecular hydrogen, which is repelled from the surface. Under space conditions, fullerenes may exist as cations, inducing a dipole moment in $H_2$, which makes physisorption of $H_2$ plausible. A recent study on amorphous carbonaceous dust structure[36] also supports the idea that atomic hydrogen is more likely to undergo chemisorption, while molecular hydrogen is more likely to be physisorbed. Understanding the mechanisms behind interstellar $H_2$ formation can improve the precision of current models and provide a more comprehensive understanding of cosmic structure evolution, in which $H_2$ plays an important role. In principle, all types of fullerene could also serve as candidates for the processes we have outlined. The mechanisms identified in this work may extend beyond fullerenes to other $sp^2$ carbonaceous dust grain surfaces.

## Methods

### Capture of atomic hydrogen by fullerene
The electronic wavefunctions are described in the plane wave basis with an energy cutoff of 25 Ry. The valence-core interactions are described by Vanderbilt ultrasoft pseudopotentials[38]. The Perdew-Burke-Ernzerhof (PBE) functional[39] is applied to compute the exchange-correlation energy. Simulation is carried out in a box size of 15 Å. The optimized structure of $C_{60}$ has a bond length of 1.40 Å and 1.45 Å, which is consistent with the experimental measurement. Since the electron energy will increase drastically during the collision, we use BOMD to ensure the electrons stay in the ground state during the propagation phase of the simulation. The simulation trajectories are integrated with a time step of 0.24 fs, with maximally localized Wannier functions[40] applied to explore the bond properties.

Extra simulations are conducted at 10K with van der Waals corrections to include physisorption. Cell size is adjusted to 20 Å and energy cutoff to 40

Ry to enable larger separation of atomic hydrogen and the fullerene. Physisorption is included with van der Waals corrections both proposed by Ambrosetti and Silvestrelli[41] and Grimme et al.[42] in NVE and NVT ensembles separately. A Nose-Hoover[43–45] thermostat operating at a characteristic frequency of 1300 cm$^{-1}$ is applied in the NVT ensemble.

### Formation of molecular hydrogen by the ER mechanism
Simulations with fullerane are carried out with a box size of 20 Å and energy cutoff of 50 Ry. Norm-conserving pseudopotentials of the Martins-Troullier type[46] are used for all atoms. Energy expectation values are calculated in reciprocal space using the Kleinman-Bylander transformation[47] for all pseudopotentials. BOMD is carried out with hydrogen atoms with various initial temperatures, ranging from 10K to 6000K.

### Formation of molecular hydrogen by sudden heating
The heating of $C_{60}H_{36}$ is simulated by CPMD because the decomposition of $C_{60}H_{36}$ takes a much longer time than the collision simulation, while the electronic structure is more stable during the heating. The fictitious electron mass is 270 au. The trajectories are generated with a time step of 0.06 fs. The heating is carried out in the canonical ensemble (NVT), with Nose-Hoover thermostat operating at characteristic frequency 1300 cm$^{-1}$. The fictitious electron kinetic energy is controlled by a Nose-Hoover thermostat operating at characteristic frequency 6000 cm$^{-1}$.

The heating of $C_{60}D_{36}$ is simulated with a fictitious electron mass of 600 au. A larger time step is allowed in this situation. The trajectories are generated with a time step of 0.1 fs. The dynamics of atoms is controlled by Nose-Hoover thermostat operating at 1300 cm$^{-1}$, while fictitious electron kinetic energy is controlled at characteristic frequency 5000 cm$^{-1}$.

### Repulsion of molecular hydrogen from adsorption
The simulation setting is the same as the capture of atomic hydrogen by fullerene. The neutral atomic hydrogen is replaced by a molecular hydrogen. Simulations are carried by initializing the temperature of incident $H_2$ from 10K to 6000K when temperature of the grain surface is set to be 10K and 50K separately. Maximally localized Wannier functions are applied to explore the bond properties.

## Data availability
Data are available from corresponding authors on reasonable request.

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

## Acknowledgements
This research was undertaken with the assistance of resources from the National Computational Infrastructure (NCI Australia), an NCRIS enabled capability supported by the Australian Government.

## Author contributions
Y.G. performed and analyzed the simulations. D.R.M. contributed to conceptualization and supervision. Y.G. and D.R.M. wrote the paper.

## Competing interests
The authors declare no competing interests.
