## [Transparent Peer Review file · Communications Chemistry]

Ab-initio dynamic study of mechanisms for dust-mediated molecular hydrogen formation in space

Corresponding Author: Professor David McKenzie

Version 0:

Reviewer comments:

Reviewer #1

(Remarks to the Author)

This paper focuses on the dust-mediated formation of molecular hydrogen in space. The authors simulate the bombardment by both atomic and molecular hydrogen of fullerene (C₆₀) and fullerane (hydrogenated fullerene C₆₀H₃₆), modeling a type of molecule that is found in molecular clouds and interstellar medium but that is also used here as a model for larger carbonaceous dust grain. The dynamics is studied with ab-initio molecular dynamics, in particular Car-Parinello molecular dynamics to stabilize the structure of the fullerene C₆₀ at the temperature of interest, followed by Born-Oppenheimer molecular dynamics to study the collisions. They look into two known formation mechanisms for molecular hydrogen formation on dust, the Eley-Rideal (direct recombination of hydrogen at the surface of the grain) and the Langmuir-Hinshelwood (through roaming of physisorbed hydrogen) mechanisms. They consider cold environments (10-100 K corresponding to molecular clouds and cold neutral medium) and warm environments (6000-10000 K corresponding to warm neutral medium, where sudden heating can happen). They emphasize on the importance of dynamics phenomena for the formation of molecular hydrogen. They conclude that fullerenes C₆₀ efficiently capture atomic hydrogen, and that at the low temperatures found in the interstellar medium, both the ER and the LH mechanisms play a role in the molecular hydrogen formation. They mention that these mechanisms may extend beyond fullerenes to other sp² carbonaceous dust grain surfaces.

The paper is well detailed in terms of methodology and parameters used to guarantee reproducibility of the results. It is well organized and written, and the work of the authors is easy to follow. The results of this work can be important to model the interstellar medium and molecular clouds, and the methodologies used can be applied to other types of dusts or colliders to study the formation of other small molecules on the surface of dust through the ER and LH mechanisms.

Nevertheless, I have some remarks and comments that should be taken into consideration as I believe they would increase the impact of this work, in particular towards the astrophysics community, to whom this study might be of interest.

1- The study focuses on fullerene and fullerane, and it is mentioned in conclusion that it might extend to other sp² carbonaceous dust surfaces. However, it is known that carbonaceous dust is more likely in the form of amorphous carbon structures (see M. Nashimoto et al 2020 ApJL 900 L40, <https://doi.org/10.3847/2041-8213/abb29d>). To better put this work and its impact in context, the introduction should mention the amorphous-type of carbonaceous dust and any insight on the proportion of carbonaceous fullerene-like dust.

2- Interstellar ice, whether it is in the form of amorphous solid water or CO, is known to act as an important catalyst in the low temperature regions of the interstellar medium, as is mentioned in the introduction. The ice covers, for temperatures of few tens of kelvins, the dust present in these environments. As the main part of this study focuses on fullerenes at around 10 K, icy mantle is very likely to be present. Dust can be partially coated by ice, hence exhibiting some surface uncovered by it, but the conclusions of this work will only apply to those uncovered area. It is important to mention this limitation in this work and possibly some insight on how much of the dust is not covered by an icy mantle at this low temperature in molecular clouds.

3- The collision dynamics were done for two cases, one where the impact site is on a carbon, and another one where it is on the ring. For those two cases, a single collision was simulated with an angle normal to the surface of the fullerene. In a realistic environment, molecular or atomic hydrogen will hit the fullerene on every possible combination of angle and position. The position, being closer to a carbon or to the ring as well as the angle, will lead to more interactions with multiple atoms of the surface of the dust before chemisorbing or physisorbing. Drawing general conclusions on the entire temperature domain considered based on a single collision at one angle for each temperature might lead to some errors on the sticking probability up to potentially 0.5 (as a collision normal to the surface will maximize the sticking at low temperature), as the sticking probability you obtain can only be 0 or 1. Some comments on this or some additional

calculations at different angles will give a more general conclusion.

4a- About molecular hydrogen not decomposing into atomic hydrogen on the surface of the fullerene, it should be mentioned that it is an expected behavior. Indeed the dissociation energy of this bond is around 48000 K, much higher than the kinetic energy of few thousands of kelvins considered in this work for the incident molecule. So there is not enough energy available to decompose the molecular hydrogen and this process is very unlikely even at few thousands of kelvins (the required energy lies at the far tail of the Maxwell-Boltzmann distribution). It is interesting to note, and might be worth mentioning in the manuscript if deemed useful, that the propensity of atomic hydrogen to efficiently stick through chemisorption (and fastly enough that it rarely has the time to physisorb) and the propensity of molecular hydrogen to physisorb is also observed on amorphous carbon dust structures (see for a study from 50 K, D. Bossion et al 2024 A&A Forthcoming article, <https://doi.org/10.1051/0004-6361/202452362>).

4b- A previous study on H₂ formation on graphite-like surface (see S. Cazaux et al 2011 A&A 535 A27, <https://doi.org/10.1051/0004-6361/201117220>) at low temperature concluded that physisorbed atomic hydrogen was forming H₂ efficiently at 15 K, graphite being relatively similar to fullerene in structure, I am curious to know if you have some thoughts or explanations on this difference of behaviour, whether it is coming from the methodology used, the surface or any other parameter. If so you should add it in your manuscript to further put your work into context.

5- The study also extends on the formation of the D₂ isotope. The D/H ratio in the interstellar medium being about 10⁻⁵, it is very unlikely that it is a molecule of interest for the astrophysical modeling (as the paper it focused toward astrophysical context). In addition, as it possesses no dipole moment, its observation and its role in molecular clouds is not as relevant as that of H₂. This comments also applies to C₆₀D₃₆, that probably cannot exist in this form in space. On the other hand, there is a great interest in the deuterium, in particular for the formation of HD. Its dipole moment makes it an efficient coolant of molecular clouds, cooling down clouds more efficiently than H₂ despite its low abundance. Studying with this formalism the collision of D on fullerene, or the collision of H on a C₆₀H₃₅D with the incoming atom hitting the fullerene in the area where the D is present would give to this work a broader impact by studying both the H₂ and HD formation on this type of dust, both very important for the astrophysics community.

6- Finally, as mentioned earlier, this work might be of importance for the astronomers and modelers of those astrophysical environments due to the role of H₂ in space, but I regret that there is no quantitative result on the H₂ formation that could be directly used in astrophysical modeling, as I believe there is in this work the material to provide so. You provide a probability of capture of atomic hydrogen, I do think that providing similarly a probability of formation of H₂ when atomic hydrogen collides with fullerene-type dust could be a valuable result to add to this work in order to make it more impactful.

I add here a few typos found in the manuscript: (p.7) "BOMD" is defined only in p.9, (p.9) "described" instead of "described", (p.9) "as used for all atoms" instead of "are used for all atoms", (p.10) "Maxially" instead of "Maximally"

Best regards,
Dr. Duncan Bossion

Reviewer #2

(Remarks to the Author)

The formation of H₂ in astronomical environments is an important topic. It has long been proposed that fullerene surfaces, in addition to dust grains, could function as catalysts. This hypothesis has received increasing attention in recent years because of the recent detections of C₆₀, C₇₀, and C₆₀⁺ in space. In this paper, through a dynamic stimulation of the interactions between H/H₂ and C₆₀, the authors report supporting evidence for the catalysis of fullerenes on H₂ formation. The results are compelling and stimulating for astrochemical research. I recommend publishing this work but only after addressing the following comments.

Abstract, 1st sentence:

Clarify what hypotheses are being tested (ER and LH mechanisms?).

P.4, 2nd para.:

C₆₀H₃₆ has a number of isomers with different symmetries. Which isomer is selected for this study? And why? What impact may investigating a different isomer have on the results?

C₆₀ and C₆₀H₃₆ have a low ionization potential and are most likely to be present as cations in space. It would be instructive to qualitatively discuss the influence of ionized species. For example, C₆₀⁺ might induce a dipole moment in H₂ and make the capture of H₂ on its surface possible.

P.4, 3rd para.:

Is there a chance that atomic hydrogen penetrates into the C₆₀ cage, forming an endo CH bond? This seems to be able to occur if the carbon cage expands by absorbing energy. If this process could be ruled out or be shown to be negligible by the current simulation, please say so. Otherwise, it ought to be incorporated in equ~2.

It should be noted that the f_C and f_R values in equ~2 as a function of temperature are displayed in Fig~S3.

P.4, last para.:

It is stated that a C-C double bond changes to a C-C single bond. C₆₀ has two types of C-C bonds, corresponding to the

hexagon-hexagon and pentagon-hexagon ring junctions, respectively. Which one is broken preferentially?

P.5, 2nd para.:

It is unclear in the last sentence what the "larger cosmic structures" refer to (molecular clouds, galaxies, or galaxy-clusters?). This sentence, in my opinion, is not really pertinent. Alternatively, the authors could mention the importance of improving the model precision for future astrochemical modelling.

P.5, 3rd para.:

Is there anything that can be said regarding the ortho-para ratio of the H₂ formed on the C₆₀ surface?

Paragraph in P.5-6:

Please provide a reference for the simulations of H₂ formation on a warm dust surface. Change "cooling...2 orders... more..." to "cooling rate...two orders...higher..."

P.6, 1st line:

It should be noted that only *small* grains experience a sudden increase in temperature.

The authors state that the collision between atomic hydrogen and C₆₀H₃₆ causes the breakage of C-H bond and transformation from sp³ toward sp² bonding (shown in Fig~2). However, this process may have a dependence on neighboring carbon atoms. Each C-H bond have 0-3 neighboring C-H bonds. Does the authors' statement hold true for all the 4 cases? If being surrounded by 3 neighboring C-H bonds, the carbon may not be transformed from sp³- to sp²-bonded.

Fig~3 is not cited in the text.

Reviewer #3

(Remarks to the Author)

This manuscript describes a novel computational study of hydrogen atom and hydrogen molecule interactions with fullerene, C₆₀. Fullerene molecules are studied both in their own right and, as the authors clearly state, as a model for surfaces of carbonaceous interstellar dust grains. Such dust grains are believed to be catalysts for molecular hydrogen formation in a wide range of environments in interstellar space. The authors carefully motivate their study with references to astronomical observations, astrophysical models as well as surface chemistry studies. Ab initio molecular dynamics calculations explore possible mechanisms for hydrogen atom adsorption at fullerenes and subsequent formation of H₂ molecules. Results indicate efficient formation of H₂ in the astrophysical environments considered. The results are highly interesting, both considering the potential role of fullerenes as promoting H₂ formation in space, as well as lessons learned about hydrogen-carbon interactions on carbonaceous dust grains in general.

The results are certainly publishable, but before publication I would ask the authors to consider and respond to the following:

p.4: The results indicate that chemisorption is a major outcome of hydrogen atom-fullerene collisions also at very low temperatures (10 K). It is stated that it was previously believed that chemisorption of hydrogen would be inefficient at such low temperatures and that H atoms were then also believed to mainly be physisorbed on dust grains.

- Here it would be very helpful to cite and discuss the references that discuss such distributions of H atoms among chemisorbed and physisorbed states.

- Could the authors also discuss what types of dust surfaces that have been considered in those previous studies. Are those surface considered to be flat and crystalline or rugged and amorphous, or both?

- Since the fullerene surface is strongly curved it would be useful to mention the expected differences in surface properties in general and H atom binding energy in particular, between fullerene and a flat graphite/graphene surface. The latter has been the topic of several previous treatments of hydrogen-carbon interactions. How does the curvature of the surface influence the binding to the surface compared to the flat surface?

- Which of a fullerene and a flat graphite surface would be the better model for a general interstellar carbonaceous grain, i.e., apart from the fullerene molecules themselves?

- The PBE density functional is used. PBE is known to not perform very well for barriers to reactions involving atomic hydrogen (or other radicals) but rather in. Could the authors cite and discuss some relevant references on barrier heights, such as to chemisorption, relevant to carbonaceous surfaces as calculated by PBE (and/or other density functionals)? How large are the uncertainties regarding the efficiency of hydrogen chemisorption because of the use of PBE (and not another density functional)?

Version 1:

Reviewer comments:

Reviewer #1

(Remarks to the Author)

The authors carefully took into consideration and replied to all the comments of the reviewers and added additional results to strengthen the conclusions of their work. I am satisfied with their answers and do not have any further comment.

Reviewer #2

(Remarks to the Author)

The authors have satisfactorily addressed all the issues raised in my earlier report. This work has been significantly improved by new simulations. I did find a few minor typos, though, which should be fixed in the final draft.

Fig.3: Please use the lower case 't' rather than 'T' to represent time since 'T' is the temperature in Eq.1.

p.10, the last line: 1300cm^{-1} -> 1300cm^{-1}

Figs. S1&S2: The units of $g(r)$ are missing. In the caption of Fig.S1, it should be stated that the values in the inset structure denote the bond length *in angstrom*.

Reviewer #3

(Remarks to the Author)

The authors have carefully responded to the comments by myself and the other reviewers. I therefore recommend publication without further review.

Detailed Response to Reviewers' comments

The reviewers' comments have been helpful in the revision of our paper. The response to each of the reviewers' comments are as follows:

To Reviewer #1:

1. **Comment:** The study focuses on fullerene and fullerane, and it is mentioned in conclusion that it might extend to other sp^2 carbonaceous dust surfaces. However, it is known that carbonaceous dust is more likely in the form of amorphous carbon structures. To better put this work and its impact in context, the introduction should mention the amorphous-type of carbonaceous dust and any insight on the proportion of carbonaceous fullerene-like dust.

Response: Amorphous carbon contains a significant fraction of sp^2 bonding, consisting of sheets with positive or negative curvature. Studies of curved graphene sheets suggests these sites locally have a strong resemblance to fullerene [1]. Additionally, substantial observational evidence for the detection of fullerene suggests that the fullerene molecule itself may be prevalent in the interstellar medium. In the revised manuscript, we have provided justification and new references for the use of fullerene as a paradigm for the study of carbonaceous dust. We have also added content about the amorphous type of carbonaceous dust in the introduction, which is highlighted in red.

2. **Comment:** As the main part of this study focuses on fullerenes at around 10 K, icy mantle is very likely to be present. Dust can be partially coated by ice, hence exhibiting some surface uncovered by it, but the conclusions of this work will only apply to those uncovered area. It is important to mention this limitation in this work and possibly some insight on how much of the dust is not covered by an icy mantle at this low temperature in molecular clouds.

Response: We thank the reviewer for raising the question of ice coverage. Recent experimental work [2] has argued that ice densities vary dramatically and that the ice coverage of dust grain in molecular clouds is less than has typically been assumed. Therefore, the existence of exposed carbonaceous dust surface is likely to be important at low temperature. When temperature increased, chemisorption of atomic hydrogen atom on the dust surface is expected to be dominant. We have added a short discussion on the impact of ice coverage in the revised manuscript highlighted in red.

3. **Comment:** In a realistic environment, molecular or atomic hydrogen will hit the fullerene on every possible combination of angle and position. The position, being closer to a carbon or to the ring as well as the angle, will lead to more interactions with multiple atoms of the surface of the dust before chemisorbing or physisorbing.

Drawing general conclusions on the entire temperature domain considered based on a single collision at one angle for each temperature might lead to some errors on the sticking probability up to potentially 0.5 (as a collision normal to the surface will maximize the sticking at low temperature), as the sticking probability you obtain can only be 0 or 1. Some comments on this or some additional calculations at different angles will give a more general conclusion.

Response: We thank the reviewers for pointing this out. To provide a more general conclusion, we conducted additional simulations with incident angles equal 30° , 60° , and 75° . We did not explore angles greater than 75° because at this angle, the hydrogen atom has already entered the range of attraction of other carbon atom and becomes captured by these nearby atoms at low temperatures.

When the impact site is a carbon atom, we tested scenarios where the temperature of incident hydrogen atom was 10K and 5500K, with 5500K being the temperature that the atom fails to be captured under normal incident conditions. When the impact site is the ring area, we tested the temperature of 10K and 6000K for the incident hydrogen atom. The simulations results show that in all cases, the hydrogen atoms are captured, as expected. Because increasing the incident angle actually decreases the velocity component normal to the surface during the collision, the atoms are more easily captured at high temperatures. At low temperature, the hydrogen atoms may either get captured directly, or interact with nearby atoms, depending on the incident angle. We have added content of additional simulations with angles in the supplementary material.

4. **Comment-a:** About molecular hydrogen not decomposing into atomic hydrogen on the surface of the fullerene, it should be mentioned that it is an expected behavior because the dissociation energy of this bond is much higher than the kinetic energy of few thousands of kelvins considered in this work. It is interesting to note, and might be worth mentioning in the manuscript if deemed useful, that the propensity of atomic hydrogen to efficiently stick through chemisorption (and fastly enough that it rarely has the time to physisorb) and the propensity of molecular hydrogen to physisorb is also observed on amorphous carbon dust structures.

Response: We thank the reviewers for pointing this out and sharing their knowledge. We have added content into the summary part of revised manuscript, which is highlighted in red.

Comment-b: Previous study on H_2 formation on graphite-like surface shows physisorbed atomic hydrogen was forming H_2 efficiently at 15K. Graphite is relatively similar to fullerene in structure, so are there thoughts or explanations this

difference of behaviour, whether it is from the methodology used, the surface or any other parameter. If so you should add it in your manuscript to further put your work into context.

Response: A previous study [3] proposes that a carbon black surface is a more suitable carbon surface for the formation of molecular hydrogen from the radiative association of atomic hydrogen. Additionally, the study with carbon black graphene sheets shows that sites with fullerene-like structures are the sites where molecular hydrogen may be formed [3]. We believe the surface structure plays a key role in this difference. Compared to planar structures like graphite, hydrogen is more likely to get captured at the bulge sites on the surface of fullerene. The study about hydrogen adsorption on graphene [4] also proposes that H atom chemisorption induces the formation of a hillock around the adsorption spot, and the main contribution to the chemisorption barrier comes from the relaxation and rehybridization of the adsorbent carbon atom directly beneath the H atom. The bulge structure of fullerene lowers the barrier, which makes chemisorption more easily to occur. We have added discussion in the revised manuscript, which is highlighted in red.

5. **Comment:** D₂ is unlikely to be a molecule of interest for astrophysical modelling, while the formation of HD is of great interest. Studying with the formalism the collision of D on fullerane will give this work a broader impact.

Response: We have conducted an extra simulation with a deuterium atom impact into C₆₀H₃₆ under 10K to explore the formation mechanism of HD under the Eley-Rideal mechanism. Simulation results indicate that both the formation of HD and H₂ are efficient on the dust grain surface. The simulation results are included in a revised Figure 2, with comparison with the formation of H₂ under the same mechanism. We thank the reviewer for this advice enabling us to further improve the work.

6. **Comment:** You provide a probability of capture of atomic hydrogen, I do think that providing similarly a probability of formation of H₂ when atomic hydrogen collides with fullerene-type dust could be a valuable result to add to this work in order to make it more impactful.

Response: We demonstrate that the formation of H₂ via the Eley-Rideal mechanism can be efficiently achieved on the surface of fullerane, with temperature of the incident atomic hydrogen ranges from 10K to 6000K. The actual probability is expected to be influenced by the temperature distribution of atomic hydrogen in the interstellar medium, as well as the concentration of hydrogen atoms, when conducting larger-scale simulations.

7. **Comment:** 'BOMD' only defined in p.9. And few spelling typos.

Response: We thank the reviewers for pointing out this for us. We have corrected all the pointed typos, marked in red in the revised manuscript. We have also defined 'BOMD' in the main text.

To Reviewer #2:

1. **Comment:** Clarify what hypotheses are being tested (ER and LH mechanisms?).

Response: We show that LH mechanisms are not the only possible mechanisms at low temperature as atomic hydrogen molecule can also get chemisorbed on fullerene-like carbonaceous dust surface, while the ER mechanism at this low temperature is also effective. Additionally, we also show the formation mechanism of molecular hydrogen under energetic events. Therefore, molecular hydrogen formation mechanisms may be a more suitable word here. We have revised the abstract to make the description more accurate, as highlighted in red.

2. **Comment:** $C_{60}H_{36}$ has a number of isomers with different symmetries. Which isomer is selected and why? What impact may investigate a different isomer have on the results?

Response: We choose one of the most common isomers of $C_{60}H_{36}$ for comparison with the experimental measurement of our calculated pair distribution function. However, we believe the choice of isomer will not affect the result because it is influenced by the structure of the impact site rather than the overall structure of the molecule.

3. **Comment:** C_{60} and $C_{60}H_{36}$ are more likely to be present as cations in space. It would be instructive to qualitatively discuss the influence of ionized species. (For example, C_{60}^+ might induce a dipole moment in H_2 and make the capture on its surface possible)

Response: C_{60}^+ has a spectrum that is remarkably consistent with several DIBs, and the capture of H_2 on its surface by inducing a dipole moment is plausible. However, this capture would be a physisorption process and would not dissociate H_2 into atomic hydrogen, meaning it will not affect our finding that carbonaceous dust assists the formation of H_2 in space. We thank the reviewer raising this point and have included additional material in the summary part of the revised manuscript.

4. **Comment:** Is there a chance that atomic hydrogen penetrates the C₆₀ cage, forming an endo CH bond? If this process could be ruled out or be shown to be negligible by the current simulation, please say so. And it should be noted that f_C and f_R values in Eq.2 as a function of temperature are displaced in Fig S3.

Response: We thank the reviewer for pointing this out. We did in fact observe this when temperature of the hydrogen atom is raised to an extremely high value. However, since in our study we focus on the cold interstellar medium, hydrogen atom is unlikely to have that high temperature under those environments. Therefore, this process is negligible in the current simulation conditions. We note that in the revised manuscript we show f_C and f_R as a function of temperature in Fig S3.

5. **Comment:** It is stated that a C-C double bond changes to a C-C single bond. C₆₀ has two types of C-C bonds, corresponding to the hexagon-hexagon and pentagon-hexagon ring junctions, respectively. Which one is broken preferentially?

Response: The structure of C₆₀ consists of alternating single and double bonds, with each carbon atom has two single bonds and one double bond. The pentagon-hexagon ring junction is broken preferentially during chemisorption of hydrogen atom. The transformation of this double bond into single bond releases energy to form CH bond. And during the formation of molecular hydrogen, the exothermic reaction of molecular hydrogen formation releases energy, which is absorbed to form C-C double bond on the impact site. That said, which one is broken preferentially depends on the reaction, while the structure of the reaction site also matters (see answer to question 10 regarding the site that is surrounded by three C-H bonds).

6. **Comment:** In the 2nd paragraph, it is unclear that in the last sentence what the “larger cosmic structures” refer to (molecular clouds, galaxies, or galaxy-clusters?). This sentence, in my opinion, is not really pertinent. Alternatively, the authors could mention the importance of improving the model precision for future astrochemical modelling.

Response: The larger cosmic structures mentioned here refer to cosmological simulations, such as those of galaxy formation. Since most of those simulations are based on rate equations or master equations, a precise study of underlying mechanisms can help establish more accurate parameters for studies that involving carbonaceous dust, especially fullerene-like dust. We have revised the sentence to make it clearer, which is highlighted in red.

7. **Comment:** Is there anything that can be said regarding the ortho-para ratio of the H₂ formed on the C₆₀ surface?

Response: In this study, we performed first principles molecular dynamics simulations that focuses on the electronic structure and the forces acting on the nuclei based on density functional theory. While of some peripheral interest, it is not possible for us to directly address this question since we do not include nuclear spin in our computational method. Additionally, the ortho-para state of the H₂ formed does not influence the mechanism we investigated. However, we note that a recent study [5] on the ortho-para ratio of H₂ on low temperature carbonaceous grains suggests that ortho-H₂ is the majority species when it forms, and the ortho-para ratio decreases as time evolves. This study [5] also indicates that bonding motifs have little effect on the nuclear spin conversion (NSC) rates, as similar results were observed on both diamond-like carbon and graphite surfaces. Therefore, the ortho-para ratio and the NSC rates should exhibit similar behavior for H₂ formed on the C₆₀ surface.

8. **Comment:** Please provide a reference for the simulations of H₂ formation on a warm dust surface. Change “cooling...2 orders...more...” to “cooling rate...two orders...higher...”.

Response: The simulations on a warm (50K) surface are also conducted in this study. We have added the capture of atomic hydrogen and the formation of molecular hydrogen on a 50K dust surface in the supplementary material. We have also changed the wording to ‘two orders of magnitude higher’, highlighted in red in the revised manuscript.

9. **Comment:** It should be noted that only ‘small’ grains experience a sudden increase in temperature.

Response: Small dust grains make up only a small amount of total mass of the interstellar medium but play a pivotal role in its chemistry [6, 7]. The sudden increase in temperature should mainly occur on small grains. We thank the reviewer for pointing this out. We have noted in the revised manuscript that the formation mechanism under sudden heat in this work is applied to small grains, which is highlighted in red.

10. **Comment:** The authors state that the collision between atomic hydrogen and C₆₀H₃₆ causes the breakage of C-H bond and transformation from sp³ toward sp² bonding (shown in Fig~2). However, this process may have a dependence on neighboring

carbon atoms. Each C-H bond have 0-3 neighboring C-H bonds. Does the authors' statement hold true for all the 4 cases? If being surrounded by 3 neighboring C-H bonds, the carbon may not be transformed from sp^3 - to sp^2 -bonded.

Response: We thank the reviewer for pointing this out. We have conducted an additional simulation involving the formation of hydrogen molecules when an incident hydrogen atom collides with a position with three neighbouring C-H bonds. In this scenario, the hydrogen will try to hover around the dust surface and may end up with, either collide with the other hydrogen atom with fewer than three neighbouring C-H bonds, facilitating the transformation from sp^3 toward sp^2 C-C bond during the collision, or become physisorbed. This new simulation highlights the importance of dust grain surface structure in the H_2 formation mechanism. We have added a short discussion of this in the revised manuscript, which is highlighted in red.

11. **Comment:** Fig 3 is not cited in the text.

Response: We thank the reviewer for pointing this out and have cited Fig 3 in the revised manuscript, which is highlighted in red.

To Reviewer #3:

1. **Comment:** The results indicate that chemisorption is a major outcome of hydrogen atom-fullerene collisions also at very low temperatures (10 K). It is stated that it was previously believed that chemisorption of hydrogen would be inefficient at such low temperatures and that H atoms were then also believed to mainly be physisorbed on dust grains. Here it would be helpful to cite and discuss distributions of H atoms among chemisorbed and physisorbed states.

Response: Previous studies suggest that physisorption dominates at low temperatures, leading to the formation of molecular hydrogen through the Langmuir-Hinshelwood mechanism. At higher temperatures, chemisorption dominates and leads to the Eley-Rideal mechanism. The separation between chemisorbed and physisorbed is determined by the temperature of the studied interstellar medium. In this study, we demonstrate that chemisorption can also occur on the fullerene surface at this low temperature in the NVE ensemble. However, as the first reviewer notes, when dust is totally covered by a thick ice layer, physisorption should still dominate. Recent experimental work [2] has argued that ice densities vary dramatically and that the ice coverage of dust grain in molecular clouds is less than has typically been assumed, which suggests that chemisorption may also be a significant process at low temperatures. Therefore, the distribution of H atoms among chemisorbed and

physiosorbed states at low very temperatures should consider the proportion of dust that is not covered by ice in future studies, rather than assuming all of it is physiosorbed. The study on amorphous carbon dust [8] also supports the propensity for atomic hydrogen to get chemisorbed and gives the temperature dependence of chemisorbed and physiosorbed states. We have added a discussion and new reference in the summary part of the revised manuscript, highlighted in red.

2. **Comment:** Could the authors also discuss what types of dust surfaces that have been considered in those previous studies. Are those surfaces considered to be flat and crystalline or rugged and amorphous, or both?

Response: The composition of dust is inferred from the observed abundances in the gas phase. Most previous studies focus on silicon and carbonaceous dust. Silicate grains in the interstellar medium are typically considered glassy or amorphous, and may be elongated or fluffy in structure. Carbonaceous dust can take various forms, such as nanodiamonds, graphite, fullerenes, kerogen [9]. The structure of grains varies depending on the specific study. In the revised manuscript, we have expanded the introduction on previous studies concerning dust grains, which is highlighted in red.

3. **Comment:** Since the fullerene surface is strongly curved it would be useful to mention the expected differences in surface properties in general and H atom binding energy in particular, between fullerene and a flat graphite/graphene surface. The latter has been the topic of several previous treatments of hydrogen-carbon interactions. How does the curvature of the surface influence the binding to the surface compared to the flat surface?

Response: A previous study with carbon black graphene sheets shows that sites with fullerene-like structures are the reaction sites where molecular hydrogen may be formed [3]. The study about hydrogen adsorption on graphene [4] proposes that H atom chemisorption induces the formation of a hillock around the adsorption spot, and the main contribution to the chemisorption barrier comes from the relaxation and rehybridization of the adsorbent carbon atom directly beneath the H atom. This supports why H atom can be chemisorbed on fullerene-like carbonaceous grain at 10K, because the structure is already curved, which lowers the energy required for chemisorption. We have added this discussion in the revised manuscript, which is highlighted in red.

4. **Comment:** Which of a fullerene and a flat graphite surface would be the better model for a general interstellar carbonaceous grain, i.e., apart from the fullerene molecules themselves?

Response: Fullerene would be a better model. Aside from the observational evidence indicating the existence of fullerene in ISM, a flat graphite surface is not easily formed in ISM. As already outlined in the text, a previous study [3] also suggests that a carbon black surface is a suitable carbon surface for the formation of molecular hydrogen, while the fullerene-like sites are where molecular hydrogen may be formed. The study of graphene [1] suggests that curved sites have locally a strong resemblance to fullerene. We have added more content about carbonaceous grain in the introduction and summary, which is highlighted red.

5. **Comment:** The PBE density functional is used. PBE is known to not perform very well for barriers to reactions involving atomic hydrogen (or other radicals) but rather in. Could the authors cite and discuss some relevant references on barrier heights, such as to chemisorption, relevant to carbonaceous surfaces as calculated by PBE (and/or other density functionals)? How large are the uncertainties regarding the efficiency of hydrogen chemisorption because of the use of PBE (and not another density functional)?

Response: PBE has been widely applied in previous studies regarding hydrogen adsorption, such as on Ni/graphene [10], metal decorated graphene systems [11], graphene [4] and graphite [12]. A comparison of hydrogen adsorption at graphene [13] using PBE-D3 and B3LYP-D3 and MP2 shows that all three methods give the same equilibrium distances, while the adsorption energies on a carbon atom is 0.015 eV lower than B3LYP-D3 and MP2. The study for pyrene+H and coronene+H systems [14] shows that the calculated barrier height from PBE is 0.05 eV lower than that from UB3LYP. Even though PBE underestimates the barrier height, the uncertainty is still reasonable comparing to the barrier height of hydrogen adsorption on graphene ~0.2 eV [4]. Comparing to B3LYP and MP2, which is extremely computationally expensive and are normally just used for structure optimization and electronic structure calculation, PBE is a better choice when exploring the dynamics.

Reference

[1] Cataldo, F. (2002). The impact of a fullerene-like concept in carbon black science. *Carbon*, 40(2), 157-162.

[2] Potapov, A., Jäger, C., & Henning, T. (2020). Ice coverage of dust grains in cold astrophysical environments. *Physical Review Letters*, 124(22), 221103.

[3] Cataldo, F. (2003). Possible Role Played by the Fullerene-like Structures of Interstellar Carbon Dust in the Formation of Molecular Hydrogen in Space. *Fullerenes, Nanotubes and Carbon Nanostructures*, 11(4), 317-331.

- [4] Ivanovskaya, V. V., Zobelli, A., Teillet-Billy, D., Rougeau, N., Sidis, V., & Briddon, P. R. (2010). Hydrogen adsorption on graphene: a first principles study. *The European Physical Journal B*, 76, 481-486.
- [5] Tsuge, M., Kouchi, A., & Watanabe, N. (2021). Measurements of ortho-to-para nuclear spin conversion of H₂ on low-temperature carbonaceous grain analogs: Diamond-like carbon and graphite. *The Astrophysical Journal*, 923(1), 71.
- [6] Burke, D. J., & Brown, W. A. (2010). Ice in space: surface science investigations of the thermal desorption of model interstellar ices on dust grain analogue surfaces. *Physical Chemistry Chemical Physics*, 12(23), 5947-5969.
- [7] Williams, D. A., & Herbst, E. (2002). It's a dusty Universe: surface science in space. *Surface Science*, 500(1-3), 823-837.
- [8] Bossion, D., Sarangi, A., Aalto, S., Esmerian, C., Hashemi, S. R., Knudsen, K. K., ... & Nyman, G. (2024). Accurate sticking coefficient calculation for carbonaceous dust growth through accretion and desorption in astrophysical environments. *Astronomy & Astrophysics*, 692, A249.
- [9] Vidali, G. (2013). H₂ formation on interstellar grains. *Chemical reviews*, 113(12), 8762-8782.
- [10] Amaya-Roncancio, S., Blanco, A. G., Linares, D. H., & Sapag, K. (2018). DFT study of hydrogen adsorption on Ni/graphene. *Applied Surface Science*, 447, 254-260.
- [11] Wong, J., Yadav, S., Tam, J., & Veer Singh, C. (2014). A van der Waals density functional theory comparison of metal decorated graphene systems for hydrogen adsorption. *Journal of Applied Physics*, 115(22).
- [12] Karlický, F., Lepetit, B., & Lemoine, D. (2014). Quantum modelling of hydrogen chemisorption on graphene and graphite. *The Journal of Chemical Physics*, 140(12).
- [13] Petrushenko, I. K., & Petrushenko, K. B. (2018). Hydrogen adsorption on graphene, hexagonal boron nitride, and graphene-like boron nitride-carbon heterostructures: A comparative theoretical study. *international journal of hydrogen energy*, 43(2), 801-808.
- [14] Wang, Y., Qian, H. J., Morokuma, K., & Irle, S. (2012). Coupled cluster and density functional theory calculations of atomic hydrogen chemisorption on pyrene and coronene as model systems for graphene hydrogenation. *The Journal of Physical Chemistry A*, 116(26), 7154-7160.

Detailed Response to Reviewers' comments

We appreciate the constructive comments and suggestions provided by the three reviewers during the review process.

To Reviewer #1:

1. **Comment:** The authors carefully took into consideration and replied to all the comments of the reviewers and added additional results to strengthen the conclusions of their work. I am satisfied with their answers and do not have any further comment.

To Reviewer #2:

1. **Comment:** The authors have satisfactorily addressed all the issues raised in my earlier report. This work has been significantly improved by new simulations. I did find a few minor typos, though, which should be fixed in the final draft.

Fig.3: Please use the lower case 't' rather than 'T' to represent time since 'T' is the temperature in Eq.1.

p.10, the last line: 1300cm^{-1} -> $1300\text{cm}^{\{-1\}}$

Figs. S1&S2: The units of $g(r)$ are missing. In the caption of Fig.S1, it should be stated that the values in the inset structure denote the bond length *in angstrom*.

Response: We thank the reviewers for pointing these out for us. We have adjusted time in Fig.3 to t. And adjust the unit to $\text{cm}^{\{-1\}}$. There is no unit for $g(r)$ because pair distribution function is dimensionless.

To Reviewer #3:

1. **Comment:** The authors have carefully responded to the comments by myself and the other reviewers. I therefore recommend publication without further review.